

# Flexor hallucis brevis motor unit behavior in response to moderate increases in rate of force development

Jeroen Aeles[1,2,3], Luke A. Kelly[1] and Andrew G. Cresswell[1]

[1] School of Human Movement and Nutrition Sciences, The University of Queensland, Brisbane, Queensland, Australia
[2] Laboratory of Functional Morphology, Department of Biology, University of Antwerp, Antwerp, Belgium
[3] Laboratory "Movement, Interactions, Performance" (EA 4334), Université de Nantes, Nantes, France

## ABSTRACT

**Background**. Studies on motor unit behaviour with varying rates of force development have focussed predominantly on comparisons between slow and ballistic (*i.e.*, very fast) contractions. It remains unclear how motor units respond to less extreme changes in rates of force development. Here, we studied a small intrinsic foot muscle, flexor hallucis brevis (FHB) where the aim was to compare motor unit discharge rates and recruitment thresholds at two rates of force development. We specifically chose to investigate relatively slow to moderate rates of force development, not ballistic, as the chosen rates are more akin to those that presumably occur during daily activity.

**Methods**. We decomposed electromyographic signals to identify motor unit action potentials obtained from indwelling fine-wire electrodes in FHB, from ten male participants. Participants performed isometric ramp-and-hold contractions from relaxed to 50% of a maximal voluntary contraction. This was done for two rates of force development; one with the ramp performed over 5 s (slow condition) and one over 2.5 s (fast condition). Recruitment thresholds and discharge rates were calculated over the ascending limb of the ramp and compared between the two ramp conditions for matched motor units. A repeated measures nested linear mixed model was used to compare these parameters statistically. A linear repeated measures correlation was used to assess any relationship between changes in recruitment threshold and mean discharge rate between the two conditions.

**Results**. A significant increase in the initial discharge rate (*i.e.*, at recruitment) in the fast (mean: $8.6 \pm 2.4$ Hz) compared to the slow (mean: $7.8 \pm 2.3$ Hz) condition ($P = 0.027$), with no changes in recruitment threshold ($P = 0.588$), mean discharge rate ($P = 0.549$) or final discharge rate ($P = 0.763$) was observed. However, we found substantial variability in motor unit responses within and between conditions. A small but significant negative correlation ($R^2 = 0.33$, $P = 0.003$) was found between the difference in recruitment threshold and the difference in mean discharge rate between the two conditions.

**Conclusion**. These findings suggest that as force increases for contractions with slower force development, increasing the initial discharge rate of recruited motor units produces the increase in rate of force development, without a change in their recruitment thresholds, mean or final discharge rate. However, an important finding was that for only moderate changes in rate of force development, as studied here, not all units respond similarly. This is different from what has been described in the literature

Corresponding author
Jeroen Aeles,
jeroen.aeles@uantwerpen.be

for ballistic contractions in other muscle groups, where all motor units respond similarly to the increase in neural drive. Changing the discharge behaviour of a small group of motor units may be sufficient in developing force at the required rate rather than having the discharge behaviour of the entire motor unit pool change equally.

## INTRODUCTION

Feet are the foundation for transmitting forces generated by the body to the ground, as well as receiving forces from the ground in its various forms. The neural control of the many muscles in and around the foot plays a particularly important role in adapting their function to many different demands (*Kelly et al., 2012*; *Kelly et al., 2019*; *Riddick, Farris & Kelly, 2019*; *Farris et al., 2019*; *Smith, Lichtwark & Kelly, 2021*). As such, one would expect neural control to be suitably adapted to allow flexible and selective activation of many of these muscles. Results from earlier studies (*Mann & Inman, 1964*; *Riddick, Farris & Kelly, 2019*) indicated that the duration and amplitude of foot muscle activation varies substantially under different conditions, suggesting a flexible muscle-force control mechanism. However, the neural control that allows such flexible adaptation of the foot has been understudied at the level of individual motor units.

One parameter that varies constantly under different task requirements is the speed at which we move. This requires the nervous system to be able to precisely regulate the contraction times and resulting force production of muscles. Previous studies have shown that the nervous system can accommodate variations in rate of force development by regulating the recruitment and discharge frequencies of motor units (*Tanji & Kato, 1973*; *Freund, Büdingen & Dietz, 1975*; *Büdingen & Freund, 1976*; *Grimby & Hannerz, 1977*; *Desmedt & Godaux, 1977a*; *Desmedt & Godaux, 1977b*; *Desmedt & Godaux, 1979*; *Duchateau & Baudry, 2014*). Collectively, these studies have found that for ballistic contractions, the recruitment threshold of motor units is lowered and their discharge frequencies increased compared to slow contractions. More recent results from simulations on the maximal rate of force development during ballistic contractions showed that the effect of motor unit recruitment was at least four times higher than the effect of the initial discharge rate (*Dideriksen, Del Vecchio & Farina, 2020*). These simulations further showed that other factors, such as the chance of doublet discharges and decreased twitch contraction times, also contribute, but to a far lesser extent than increased motor unit recruitment (*Dideriksen, Del Vecchio & Farina, 2020*). However, this collective knowledge is based on the comparison between slow and very fast ballistic contractions. The large difference in contraction speeds for these studies leaves unanswered questions remaining about the neural strategies for increasing the rate of force development at more moderate contraction speeds, creating a significant knowledge gap given that moderate contraction speeds are likely closer and more relevant to most daily activities.

In previous work, we have shown that the flexor hallucis brevis (FHB), a small intrinsic foot muscle, has a large range of motor unit recruitment thresholds (up to 98% of maximum voluntary contraction (MVC)) as well as a substantial range of discharge frequencies (4.1–34.2 Hz) during slow isometric contractions (*Aeles et al., 2020*). This is in contrast with other similarly sized muscles, such as those found in the human hand, that usually have high discharge frequencies, but only recruit motor units up to moderate force levels, *i.e.,* up to 50% and 61% of MVC for the adductor pollicis and the first dorsal interosseus, respectively (*Duchateau & Hainaut, 1981*; *Moritz et al., 2005*). We speculate that the different discharge behavior in FHB is due to the requirement for it to produce forces for very different conditions, *i.e.,* slow tasks such as controlling standing balance to more rapid tasks such as required during push-off in walking, running, and jumping. As opposed to other intrinsic foot muscles, FHB is one of few muscles that does not span the longitudinal arch. Its sole function is to flex the big toe, thereby allowing its activation to be separate from the other foot muscles that, besides their primary function, also regulate arch stiffness. FHB is therefore a foot muscle suitable to study the behavior of individual motor units during isolated contractions at different rates of force development.

Our knowledge on how the motor units of the intrinsic foot muscles behave under varying rates of force development is limited. Only two studies have investigated the motor unit behavior in these muscles under such conditions (*Grimby & Hannerz, 1977*; *Grimby, Hannerz & Hedman, 1979*). In these studies, Grimby and colleagues showed increasing discharge frequencies with faster toe extension movements. However, their exploratory study used movements that were less controlled, making it hard to make precise deductions from the results in terms of changes in motor unit recruitment thresholds and discharge frequencies. To our knowledge, no study has systematically investigated the changes in recruitment threshold and discharge rate of motor units of intrinsic foot muscles with changes in the rate of force development during contractions at speeds similar to what occurs during daily living.

In this study we therefore compared the recruitment threshold and discharge frequencies of single motor units of an intrinsic foot muscle, FHB, in an isometric toe flexion task performed at two different, non-ballistic, rates of force development. Based on previous research in other lower limb muscles at varying rates of force development, we hypothesized that the recruitment threshold would decrease, and the discharge frequencies increase with increasing rate of force development. The data used in this study was collected as part of a larger experiment, for which parts have been published in *Aeles et al. (2020)*.

## MATERIALS & METHODS

### Participants

Ten male participants (mean age: 30 ± 6 yrs, mass 80 ± 10 kg, height 180 ± 4 cm) volunteered to participate in this study. Females were also invited to participate in the study, but none volunteered. Participants were healthy with no musculoskeletal pain in the previous three months and had not been diagnosed with any neurological disorders. Participants were asked not to participate in vigorous exercise one day prior to and on

the day of the experiment and to abstain from alcohol consumption on the day of the experiment. They were allowed their normal levels of caffeine. The project was approved by the Human Research Ethics Committee of The University of Queensland and conformed to the Declaration of Helsinki with the exception of study registration (ethics approval number: 2018000460). All participants provided their written informed consent prior to familiarization and data collection.

## Experimental set-up and protocol

Eight participants were tested on three different sessions with seven days in between. This was done to obtain a greater sample of decomposed motor units. The two other participants were only tested during a single session due to their time constraints. During some sessions, no distinct motor unit data could be decomposed. As such, for five participants, all data came from a single session, while the data from the remaining five participants came from multiple sessions. Participants were not all tested at the same time of the day.

Participants were seated on a chair with their right foot on a custom-built platform (Fig. 1), equipped with a load cell (~1,950 N, 2 mV/V, DACELL, KR). Using a previously described approach (*Aeles et al., 2020*), participants were asked to perform an isometric metatarso-phalangeal (MTP) joint flexion task of the right big toe against the load cell, which recorded the force produced during big toe flexion. The load cell data were amplified 1,000 times using a custom-built amplifier and analogue-to-digital converted at 4 kHz (Spike2 & Micro3–1401; Cambridge Electronic Design, UK). Participants were asked not to flex the interphalangeal joint to minimize potential co-activation of the flexor hallucis longus (FHL) muscle. To allow a comparative interpretation of the joint moments with other studies and tasks, calibration measurements were made for two participants. Participant 1 applied a maximal force of 97.6 N with an external moment arm of 4.7 cm, resulting in a MTP flexor moment of 4.6 Nm (*i.e.,* 2.3 Nm = 50% MVC as used in this study). Participant 2 applied a maximal force of 78.4 N, with an external moment arm of 5.2 cm, resulting in a MTP flexor moment pf 4.1 Nm (*i.e.,* 2.1 Nm = 50% MVC as used in this study). The foot was secured in a consistent orientation with brackets located at the heel and ankle. The knee and hip joint were positioned at approximately 90° of flexion, while the left leg rested next to the platform. All participants were extensively trained to perform the submaximal MTP flexion task at two different ramp rates over a four-week period prior to the data collection sessions. This training period was needed for participants to adequately perform the MTP flexion task in the absence of inter-phalangeal joint flexion. This task was designed to specifically isolate the force produced by FHB that inserts into the proximal phalanx, while minimizing any contribution from the FHL muscle, that inserts on the distal phalanx of the hallux. Participants performed three training sessions per week consisting of the same ramp contractions as detailed below up to 50% and 100% of MVC, until participants were able to reliably perform the task (as assessed by surface EMG of FHB and FHL and their force-matching ability). During the MVC trials, participants were asked to perform the task to a maximal effort level and were asked to reach peak force in approximately 2 s and to hold for another 2 s, to avoid performing a rapid ballistic contraction. Each participant also performed the task at 125% of MVC force, prioritizing

force over task accuracy, for which the surface EMG amplitude of FHB was compared to the 100% MVC trials. If no further increase in FHB surface EMG amplitude was visible, it was deemed that the participant was maximally activating the muscle. Some participants only required one or two training sessions, while others required three to four weeks of training. Data from the training sessions was not recorded. The MTP flexion MVC used for the experimental sessions was determined during the final training session and used to determine the target for the ensuing data collection sessions. For this assessment, three MVCs were recorded with 90-s rest between each contraction.

Before all sessions, participants performed a standardized warm-up, consisting of a series of FHB ramp and hold contractions with force plateaus ranging from 20% to 50% of MVC. None of the participants reported any signs of fatigue after this warm-up, which also served as a brief re-familiarization to the task Following the warm-up, a single MVC was performed, and the peak force was compared to the pre-determined MVC obtained during the training sessions. The MVC obtained during the final training session and each experimental session closely matched for all participants. During the data collection sessions, participants were presented with multiple sets of trapezoidal target force ramps on a display in front of them (Spike2, Cambridge Electronic Design, UK). These ramps were categorized as either slow or fast according to the time-to-plateau, which was set at either 5 s or 2.5 s, respectively. During these times, participants increased their MTP force from 0 to 50% of MVC, which results in an MTP flexor moment of approximately 2.2 Nm). Participants were asked to hold the target force at 50% for 3 s before following a decline of the target ramp over 5 s or 2.5 s. A single set consisted of both a slow and fast ramp, with a minimum of 30-s rest between the two ramps, during which the participants were instructed to completely relax to avoid any movement of the indwelling wires. The order in which the ramps were presented within a set was quasi-randomized so that all participants started some sets with the slow and some sets with the fast conditions. Participants were given 60-s of rest following each set depending on the quality of the indwelling EMG signal, between 2 and 5 sets were completed during a session. Data for another study was collected during the same experimental session (see *Aeles et al., 2020*). This consisted of multiple ramp contractions under similar conditions up to 100% of MVC, with 90 s of rest following each contraction. The order of these trials was also randomized and mixed with the slow and fast sets, with a maximum of two consecutive MVCs. The total number of contractions was less than the participants performed during the training period and together with the chosen randomization protocol, we expected no consistent effect of fatigue. The position of the fine-wire electrodes was slightly adjusted between sets by withdrawing the electrodes approximately 1 mm, allowing data to be collected from a different region of the muscle.

## EMG recordings

Quadrifilar fine-wire EMG electrodes (304; California Fine Wire Company, Grover Beach, CA, USA) were fabricated and inserted into the medial head of FHB under ultrasound guidance (Ultrasonix L14-5/38; Sonix MDP, Peabody, MA, USA) using a 25-gauge needle (0.50 mm diameter, 38 mm length). Ultrasound guidance was required because of the small size of the muscle, its location and to avoid damaging any nerves or vessels in proximity.

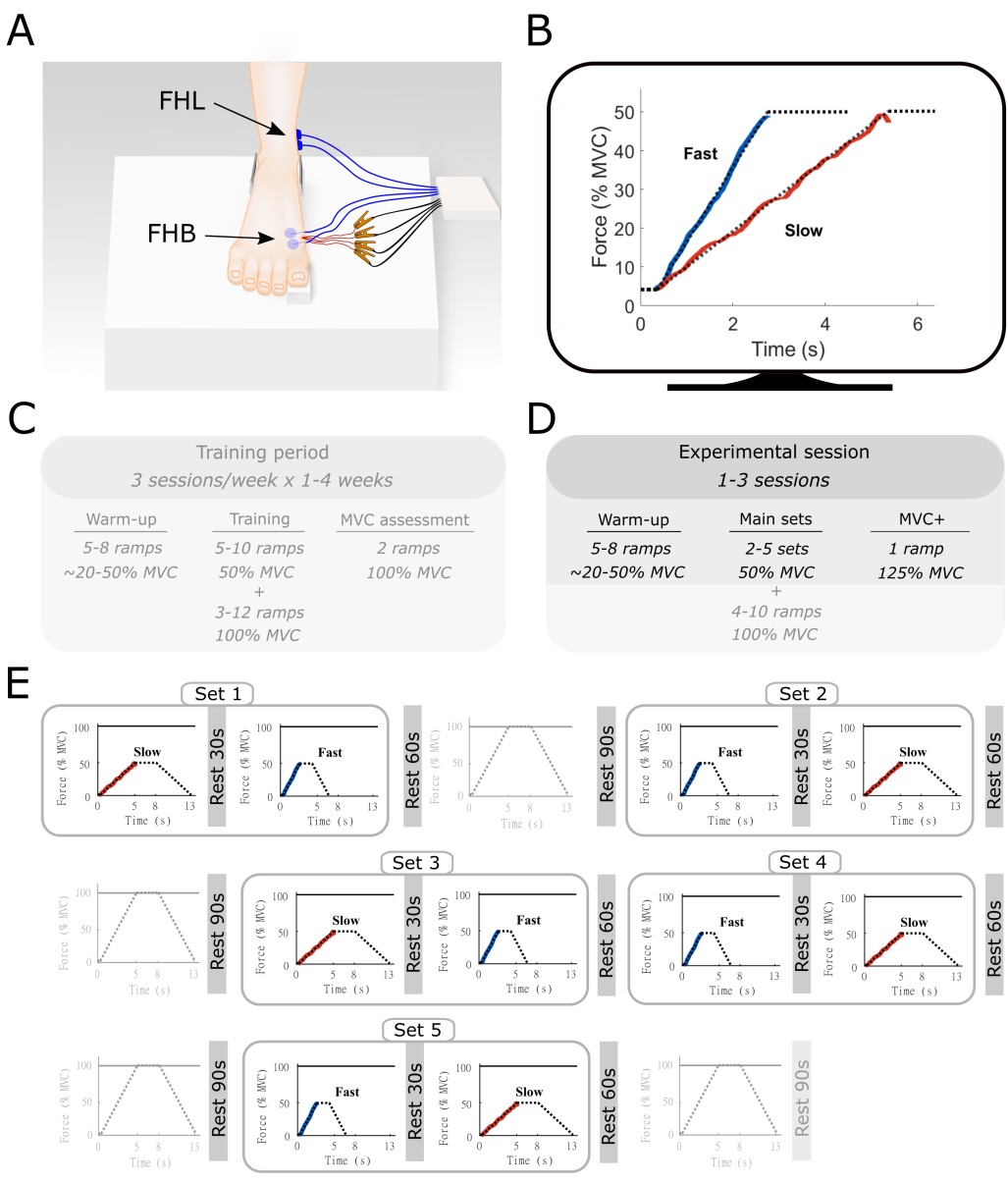

**Figure 1** **Methods overview figure showing the experimental setup as well as an example of the force ramp participants matched.** Experimental setup (A) with example data from the force-matching task (B), and the experimental protocol (C–E). (A) the right foot of the participant was placed on a custom-built platform and secured near the ankle. The big toe was placed on a small wedge that was connected to a force transducer underneath. The location of the surface electromyography electrodes for flexor hallucis brevis (FHB) and flexor hallucis longus (FHL) are shown in blue and highlighted with the arrows. The indwelling fine-wire electrodes into the FHB muscle belly are also shown in red. (B) an example of the target force traces (dotted lines) for the fast and slow ramp conditions. 

**Figure 1 (…continued)**
The actual force output from one participant is shown in blue (fast) and red (slow). While participants followed the entire ramp and hold, only the ascending limb of the force curve was used for analyses and thus only this part is shown for clarity. The ramps were shown individually on a monitor in front of the participant. (C) an example overview of the training period that participants followed prior to the start of the study. This training was done in order to allow participants to perform the task properly. Data from the training was not recorded and therefore not used in this study. (D) Overview of the experimental sessions. Note that additional ramps up to 100% MVC were performed as part of a greater study. All aspects that were not part of the current manuscript are made transparent (E) Example of a single experimental session. Sets consisted of one slow and one fast ramp with rest in between. Several sets were performed in the same experiment session and were ordered quasi-randomly. Ramps up to 100% MVC were not analyzed as part of the current manuscript and are therefore made transparent.

The needle was inserted from the medial side of the foot, thus penetrating the distal compartment of the abductor hallucis muscle. When the tip of the needle was close to the lateral border of FHB, the needle was carefully retracted, leaving the wire electrodes in place. The lateral border of the muscle was chosen as target location because it was the furthest away from where the wires were inserted into the foot, and we could thus retract the wires to record from slightly different areas in the muscle when required. This retraction of the wires was done when no motor unit was clearly visible during a 50% MVC contraction or after recording at least one trial. By retracting the wires minimally, we could often pick up clearly distinguishable action potentials from motor units that went previously undetected. During the ultrasound imaging, it was noted how far the wires could be retracted for each individual before the wires would move outside of FHB and into the abductor hallucis, at which point the experimental session was ended. Each wire had a diameter of 25.4 $\mu$m, with only the tip of the wires exposed as the recording area. Using a medical-grade ethyl cyanoacrylate glue (Cyberbond 2241, Engineering Adhesives & Lubricants Pty Ltd, AU), we glued the four wires together. From the electrodes, we recorded two bipolar channels of intramuscular EMG data. The data were amplified 1,000 times, analogue filtered using a bandwidth filter between 50 Hz and 5 kHz (Neurolog NL900D; Digitimer, Fort Lauderdale, FL, USA) and then analogue-to-digital converted at a sampling rate of 20 kHz (Spike2 & Micro3 –1401; Cambridge Electronic Design, Cambridge, UK). Bipolar surface EMG electrodes were placed on the skin with an inter-electrode distance of 20 mm over the FHB and FHL, under ultrasound guidance (*Péter et al., 2015*). A ground electrode was placed on the lateral malleolus. Surface EMG data were amplified 1,000 times before being analogue high-pass filtered at 10 Hz and were then analogue-to-digital converted at a sampling rate of 4 kHz using the same software and equipment as were used for the fine-wire EMG data.

## Motor unit decomposition

Only data collected from the ascending limb of the force ramp was analyzed. The recordings from the intramuscular EMG were then semi-automatically decomposed into single motor unit action potential trains based on their shape and amplitude (Spike2 software, Cambridge Electronic Design, UK). Only trials that had identifiable motor unit action potentials across the entire ramp were used. If the same motor unit was visible in multiple trials, the trial with the smallest force root mean square error (RMS, detailed below) was used. Each identified action potential was then visually checked, based on discharge rate and shape, and corrected

if it was attributed to the wrong motor unit. Both channels of intramuscular EMG data were decomposed. To reduce any potential discrimination bias, all data was decomposed twice on two different occasions, with at least seven days between decompositions of the same trial by the same investigator, in a random and blinded manner. Trials that did not yield matching results between these two decomposition attempts were excluded from further analyses. Only motor units within a single set for which the action potential trains could be decomposed from recruitment to the end of the ascending limb of the ramp in both fast and slow trials were used for further analysis. A representative example of the decomposition results and action potential templates for the fast and slow conditions from one set are shown in Fig. 2. We used both a qualitative and quantitative assessment of how similar the shape of the action potentials of a single motor unit were, both within a single trial as well as between the slow and fast trials within a single set. For the quantitative approach we only used the motor units for which action potentials were successfully discriminated by the Spike2 algorithm. Some motor units were fully discriminated manually and therefore not included in this analysis. To discard any voltage offsets between the action potentials the value of the first data point was subtracted from each subsequent data point of the same action potential. A normalized cross-correlation analysis on all action potentials within a single trial for both the fast and slow conditions was performed. The cross-correlation method compared the shape of a single offset-removed action potential with the shape of the next offset-removed action potential by shifting the latter with a lag that was set between $-10$ and $+10$ data points. The mean of the peak correlation values of each of the comparisons within a single trial was then calculated to obtain a single value as a measure of action potential similarity. The same approach was then used to compare the action potentials between the fast and slow trial within a set for each motor unit.

The target ramp and force data as well as the time stamps of the motor unit discharges were imported to Matlab (R2018b, The Mathworks, MA, USA) and time synchronized. Force values were normalized to the respective MVC, and the RMS calculated using the target ramp and force data. The standard deviation (SD) of the force data was calculated after removal of the linear trend. Motor unit discharge rate were derived from the inter-spike intervals determined from the absolute time stamps of each discharge. Subsequently, all data was resampled to contain 50 data points per second (50 Hz), prior to low-pass filtering at 0.5 Hz with padding applied before and after the data. Any discharge frequencies lower than 4 Hz or higher than 50 Hz were excluded, as suggested by *Moritz et al. (2005)*. The initial, final, and mean discharge rate of each motor unit recording were calculated. The initial discharge rate was defined as the discharge rate at recruitment, the final discharge as the last discharge of the ascending limb of the ramp. The coefficient of variation of a 0.5-s window was calculated from the discharge rate data before moving the window across the data in steps of 1 ms. When the coefficient of variation in the window was lower than 50%, the absolute time stamp of the first motor unit discharge in this window was used to find the force value at that time instant. This force value was then defined as the recruitment threshold of that motor unit (*Moritz et al., 2005*).

All the recorded FHB and FHL surface EMG data were band-pass filtered offline (20–400 Hz) using a fourth-order Butterworth filter, then rectified and low-pass filtered at 5 Hz

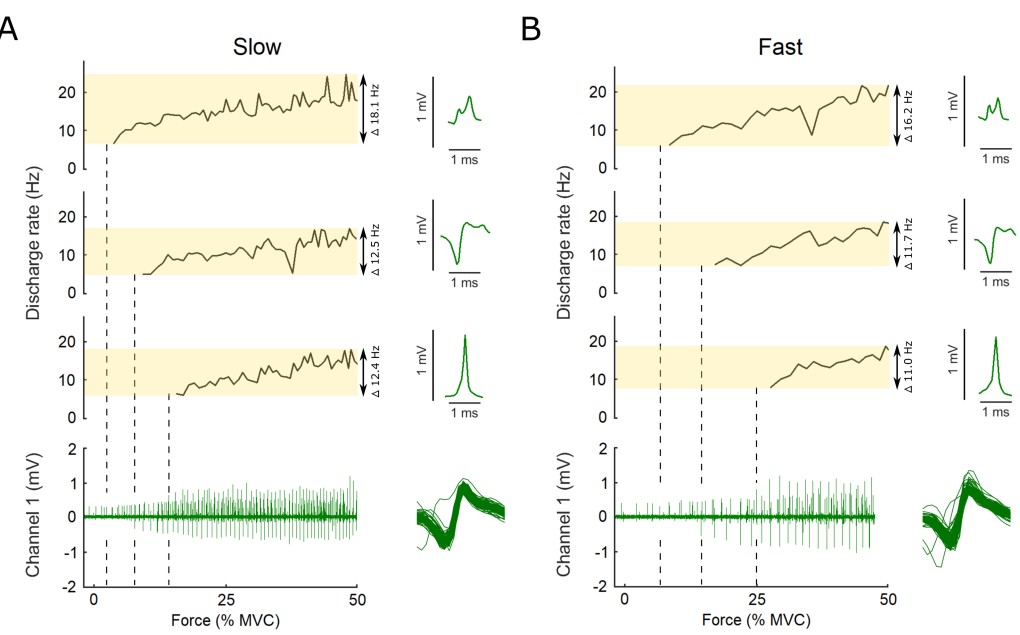

**Figure 2 Example data, the raw EMG recording is shown for both the slow and the fast conditions as well as the action potential template for three motor units.** Examples of the raw data of the indwelling fine-wire recording (bottom), discharge rates (top) and action potential templates (right) of three motor units that were decomposed from an intramuscular EMG recording from flexor hallucis brevis over the ascending limb of the force ramp in both the slow (A) and fast (B) conditions. Note the *x*-axis is presented here as percent of maximal voluntary contraction, not time, which is different for the slow and fast ramps (*i.e.,* 5 s to 50% MVC for slow and 2.5 s to 50% MVC for fast). Vertical dashed lines show the first action potential of each unit on the EMG trace. The variation between units and conditions is highlighted by the transparent yellow rectangles and the range in discharge rates is reported. Overlaid action potentials are shown for an example motor unit. There is some variation expected for some firings due to superimposition of action potentials of other motor units. During analysis, subtracting these waveforms from the original confirmed that they belonged to the motor unit shown as example here.

using a fourth-order Butterworth filter. The data of each muscle were then normalized to the peak value from the MVC trials. In a last step, the RMS amplitude for this normalized surface EMG data of FHB and FHL was computed across the entire ramp. Due to technical issues with some of the surface EMG data, only 8 participants for FHB and 9 participants for FHL were used for statistical comparison between the fast and slow ramp.

## Statistics

All statistical analyses were performed in Matlab version R2018b. A repeated measures nested linear mixed model was constructed for the input parameters: recruitment thresholds, the initial, final, and mean discharge rate, the number of action potentials and the time they were discharging (*Tenan, Marti & Griffin, 2014*). We used the conditions (fast and slow) as the fixed effect in the model while random slopes were introduced with the participant that a respective motor unit belonged to as random effects. The model equation had the form:

$$y \sim 1 + \text{Cond} + (1|\text{Part}:\text{MUnr}) + (1|\text{Part}).$$

With y denoting one of the input parameters (*e.g.*, recruitment thresholds), Cond the two conditions, Part the participant and MUnr the motor unit. These parameters were then individually compared between the slow and fast condition using Satterthwaite's method for obtaining degrees of freedom for $F$-tests. The RMS amplitude for the surface EMG of FHB and FHL, and the force RMS error and the de-trended force standard deviation, of the slow and fast trials within a single set were compared with Student's $t$-test. A linear repeated measures correlation was computed between the difference in recruitment threshold and the difference in mean discharge rate between the fast and slow ramp (*Marusich & Bakdash, 2021*). This type of correlation takes into account repeated measures on a subset of participants and accounts for differences in the number of motor units per participant included in the analysis. Significant levels for all statistical tests were set at $\alpha \leq 0.05$.

## RESULTS

The majority of the intramuscular recordings were highly selective, with motor units being clearly distinguishable to the target force of 50% MVC. Typically, between one and three motor units were identified within a single set. In total, we obtained action potential trains from 33 motor units that were matched within a single set between the slow and fast conditions. The shape and amplitude of the action potentials were highly consistent within and between trials of a single set. The cross-correlation analysis for the slow condition and between the slow and fast conditions was performed on the action potentials from 19 motor units while the analysis for the fast condition was performed on 21 motor units. A very high mean cross-correlation value was found for the slow and the fast conditions separately ($R^2 = 0.80$ for both), with the lowest and highest value found in a single motor unit equal to $R^2 = 0.37$ and $R^2 = 0.96$ in the slow and $R^2 = 0.42$ and $R^2 = 0.98$ in the fast condition, respectively. A high mean cross-correlation value was found between the action potentials of the slow and the fast conditions for each motor unit ($R^2 = 0.76$), with the lowest and highest value found in a single motor unit equal to $R^2 = 0.37$ and $R^2 = 0.98$. The action potentials of the remaining motor units were assessed qualitatively and were all highly distinctive as seen in the example action potential waveforms on Fig. 2. All further analyses were performed on all 33 motor units.

The minimum number of action potentials per motor unit during the ramp was six for the fast and five for the slow conditions. The total number of action potentials for a single motor unit during the ramp was significantly greater in the slow compared to the fast conditions ($38 \pm 22$ *versus* $19 \pm 9$ respectively, $P < 0.001$). The average time that motor units were discharging action potentials during the 5 and 2.5 s slope was significantly different for the slow and fast conditions, $2.78 \pm 1.37$ s, and $1.32 \pm 0.52$ s, respectively ($P < 0.001$).

The recruitment and discharge behavior varied substantially between individual motor units from the same participant as well as between participants (Fig. 3). Recruitment thresholds were not significantly different between the fast and slow conditions, with mean thresholds of $24 \pm 11$% MVC (range: 6–43% MVC) for the slow and $23 \pm 10$% MVC

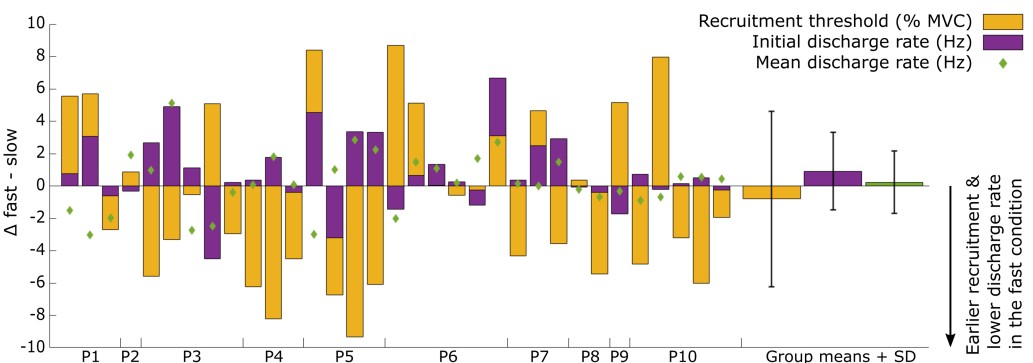

**Figure 3 Overview of the individual motor unit responses to a change in rate of force development.**
Changes in recruitment threshold (yellow bars), initial and mean discharge rate (purple bars and green diamonds respectively) between the fast and slow conditions. The $y$-axis represents the absolute values of the change in each respective parameter and unit (% MVC for recruitment threshold and Hz for the discharge rate). The direction of the change is presented on the bottom right side of the graph. Each bar represents a single motor unit and the units are grouped per participant (P1-10). The group means and standard deviation for the three variables are also provided as bars and error bars. Large variability between motor units in all three parameters can be observed from this figure, not uncommonly for motor units from the same participant.

(range: 5–38% MVC) for the fast conditions ($P = 0.588$). The discharge rate of all motor units ranged from 4.5 to 31.8 Hz in the slow condition and 4.1–26.3 Hz in the fast. The initial discharge rate was approximately 10% slower for the slow than the fast condition (mean: 7.8 ± 2.3 Hz *versus* 8.6 ± 2.4 Hz, respectively, $P = 0.027$). No difference was found between the two conditions for the mean (13.2 ± 3.1 Hz *versus* 13.4 ± 3.1 Hz for slow and fast respectively; $P = 0.549$) or final (15.4 ± 5.3 Hz *versus* 15.1 ± 4.3 Hz for slow and fast, respectively; $P = 0.763$) discharge rate. A significant but small negative correlation ($R^2 = 0.33$, $P = 0.003$) was found between the difference in recruitment threshold and the difference in mean discharge rate between the two conditions (Fig. 4). This coefficient can be interpreted as the fit for the common relationship between these parameters among the participants.

There was no significant difference between the two ramps in RMS amplitude of the surface EMG for FHB (59 ± 19% MVC *versus* 60 ± 19% MVC for fast and slow respectively) nor FHL (18 ± 8% MVC *versus* 18 ± 8% MVC for fast and slow respectively). Also, the ability to match the target force was similar between the two ramp conditions (mean RMS error: 2.4 ± 1.0% MVC and 2.2 ± 1.4% MVC for fast and slow respectively). Although the force variability was low in both conditions, it was significantly lower in the slow (standard deviation: 1.1 ± 0.3% MVC) compared to the fast (standard deviation: 1.3 ± 0.3% MVC) condition.

## DISCUSSION

In the current study we compared the motor unit discharge properties of FHB during two ramp contractions with different, but not extreme, rates of force development. The only

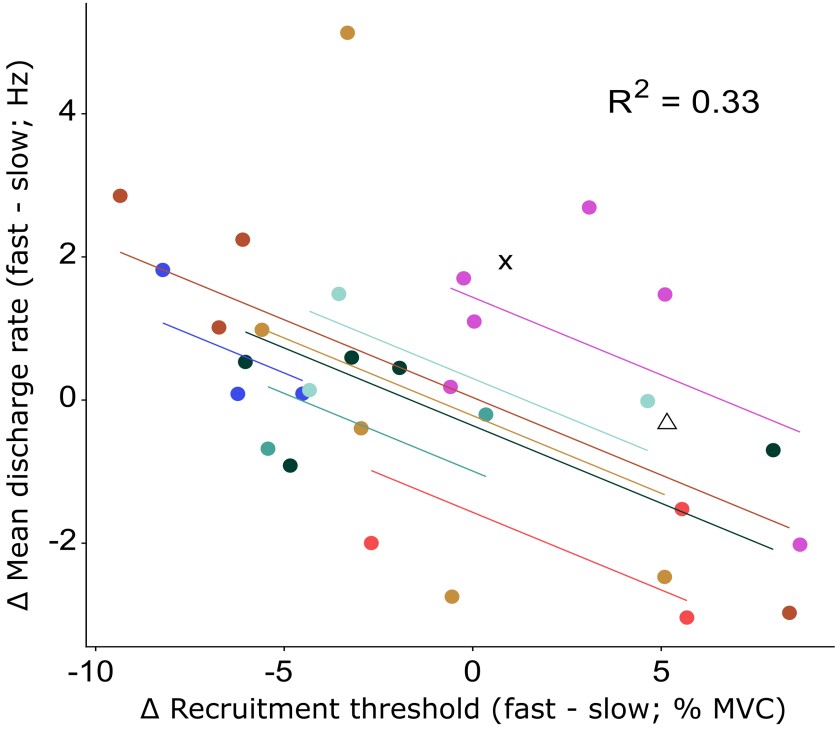

**Figure 4** **Repeated measures correlation results figure with all data points.** Relationship between the change in recruitment threshold and the change in mean discharge rate between the fast and slow condition. The motor units of each participant and the least-squares lines are grouped per color. The motor units of two participants for which only a single motor unit was analyzed are shown by the cross and triangle respectively and were not included in the repeated measures correlation analysis. The correlation coefficient represents the fit for the common regression slope for the data, or the common relationship among the participants. The coefficient here shows a statistically significant, but low correlation.

significant difference between the two ramp contractions at the group level was found in the initial discharge rate of the motor units.

Contrary to studies that compared slow to very fast or ballistic contractions, we found no change in the recruitment threshold of matched motor units. We also found no significant increase in surface EMG amplitude for the faster contraction, which suggests that there was no large increase in the number of motor units recruited for the faster contractions. These results show that despite the two-fold increase in rate of force development (from 10% to 20% MVC/s), only minimal compensations in motor unit discharge behavior occur in this relatively small muscle which is typically used during locomotion for forward propulsion (*Farris et al., 2019*) and postural control during standing. However, there was substantial variability in the changes in individual motor units, suggesting that for moderate changes in rate of force development, only some but not all motor units change their firing behavior (Fig. 3). The increase in initial discharge rate observed here appears sufficient to accommodate the increase in rate of force development compared to when all motor units are affected similarly, such as occurs when comparing slow to ballistic movements.

Increasing the discharge rate at which motor units start to discharge during a fast compared to slower muscle contraction results in the force output increasing more rapidly (*Parmiggiani & Stein, 1981*). Use of this mechanism to rapidly increase force has also been reported in other studies (*Grimby & Hannerz, 1977*; *Desmedt & Godaux, 1977a*). However, despite the higher initial discharge rate, the overall mean and final discharge rates in our study were similar between the two conditions. It has previously been reported in other muscles that during ballistic contractions, motor units initially fire at very high frequencies for the first discharges (*Duchateau & Baudry, 2014*; *Desmedt & Godaux, 1977a*; *Desmedt & Godaux, 1977b*; *Desmedt & Godaux, 1979*), after which the discharge rate is significantly lowered. The difference in initial discharge rate between the slow and fast condition in our study are of much smaller magnitude than those previously reported and may be explained by the smaller difference in rates of force development, which are presumably more comparable to rates of force development observed during daily activities, compared to previous studies, which only considered ballistic contractions (*Duchateau & Baudry, 2014*; *Desmedt & Godaux, 1977a*; *Desmedt & Godaux, 1977b*; *Desmedt & Godaux, 1979*). It is possible, given these findings, that the relative importance of increasing the initial discharge is somewhat higher during low rates of force development compared to ballistic contractions. This may explain why we found no difference in the mean or final discharge rate, although this is mostly speculation.

One aspect that needs to be considered is that several participants in our study needed an extensive period of training before they were able to consistently perform the task. While all participants reached an adequate level of performance prior to the experimental sessions, it is possible that our findings do not reflect a well-trained neuromuscular system. Short-term training has been found to cause changes in motor unit firing behavior such as changes in the discharge rate (*Duchateau, Semmler & Enoka, 2006*; *Christie & Kamen, 2010*), and is dependent on the type of training (*Martinez-Valdez et al., 2017*). It is currently unclear if the same findings would have been observed for untrained participants. However, it must be emphasized that the training in this study was solely intended to familiarize the participants with the task and to ensure that it was performed properly, specifically with consistent contribution of the FHB muscle and with minimal contributions from other toe flexors. It also needs to be considered that the findings are extracted from a relatively small pool of motor units. There is a possibility that with more motor units included, their discharge behavior would be less variable, although we believe this is unlikely since we increased the size and representation of the entire motor unit pool by measuring several participants in multiple sessions, and by retracting the wires slightly between sets. Both these measures increase the chance that motor units from slightly different regions of the muscle were included. While we did not control for different caffeine ingestions, state of arousal, sleep, or different time of day between measurement sessions, it is unlikely that these would have systematically affected our findings. However, we cannot exclude the possibility that they were confounding factors for the variability between motor units.

Although both of our ramp conditions could be considered to be relatively slow, there was still a twofold increase in the rate of force development between the two conditions. As we found no difference between conditions in motor unit recruitment threshold, and
mean and final discharge rate, it is surprising that a minor increase in initial discharge rate was sufficient to produce the required increase in rate of force development. While *Desmedt & Godaux (1977a)* mostly focused on discussing the comparison between a slow and ballistic contraction, their results do not show any significant group changes in the recruitment threshold of motor units between two slow contractions, where the one contraction also had a twofold increase in the rate of force development compared to the other slow contraction (Fig. 3A in (*Desmedt & Godaux, 1977a*)). Since group averages of the change in initial discharge rate were not reported, it is not possible to determine from their study whether changes in motor unit discharge behavior occurred in the faster of the two muscle contractions. However, based on these combined findings we could speculate that when the rate of force development needs to be increased at relatively slow contraction speeds, a change in initial discharge rate may be sufficient to achieve the required greater rate of force development.

While the use of highly selective indwelling fine-wire electrodes is beneficial for accurately following the same motor units across conditions, their limitation is that only a few motor units are generally recorded in each participant. Other motor units within the muscle, which may have behaved similarly or differently, have perhaps gone undetected. However, we also found no increase in the amplitude of the surface EMG signal in either FHB or FHL. While no exact information on motor unit recruitment can be extracted from the surface EMG, it suggests at least that there was no large difference in the number of motor units that were recruited between the two conditions for FHB and that there was no greater increase in muscle activation in the agonistic muscle, FHL. Our pilot data (not presented here) suggested also no significant contribution of any of the triceps surae muscles at either of these rates of force development and torque levels. While there are limitations to the use of surface EMG for measuring FHL activation (*Péter et al., 2019*), these limitations are assumed to mostly arise from muscle–tendon mechanics during active joint rotations, which did not occur in our experimental setup. However, it is possible that the muscle activity using surface EMG in our experiment underestimated the true muscle activation, yet it is likely that this occurred to a similar extent in both the fast and the slow condition. Another possibility for the lack of significant differences may be due to low statistical power because of the relatively low number of motor units. Based on previous findings (*Aeles et al., 2020*) and the number of motor units in another foot muscle, abductor hallucis (*Johns & Fuglevand, 2011*), we speculate that the total number of motor units in this small muscle, FHB, is low, only allowing for a few motor units to be detected, especially since they need to be detectable in both conditions. However, contrary to what is expected during ballistic contractions, it is more likely that not all motor units are required to respond similarly to a less extreme increase in rate of force development.

We found a large range of motor unit responses both within and between conditions (Fig. 3), which indicates that motor units within the same muscle can present different discharge behavior even though the task, *i.e.,* producing a flexion torque, albeit with different rates of force development, appear to be similar, and well controlled. It is possible that some or all of this variability in motor unit behavior is driven by variable activation of other toe flexors such as the abductor hallucis (*Kelly, Racinais & Cresswell, 2013*). We

were not able to simultaneously record EMG for the abductor hallucis and FHB during the experiment due to hardware limitations. Regardless, our findings show that, as opposed to ballistic contractions, for moderate increases in rate of force development, not all motor units change their firing behavior similarly. The negative correlation between the change in mean discharge rate and the change in recruitment threshold suggests that motor units whose mean discharge rates increased more from slow to fast, also have their recruitment threshold lowered. This indicates a greater emphasis by the nervous system on changing the discharge behavior of specific motor units, when the rate of the generated muscle force only needs to increase moderately. By lowering the recruitment threshold of several larger motor units while also increasing their initial discharge rate, it is possible that changes in other, perhaps smaller motor units that innervate less fibers and thus affect the force less are not required lest the force be increased too quickly, resulting in an overshoot from the target force. By adapting such a strategy, the nervous system would still have other motor units to recruit whose discharge behavior could also be increased when even faster muscle contractions are required. We speculate that such a mechanism would provide more flexibility in force generation capability compared to a mechanism where all motor units always respond similarly, which appears to not be required at lower rates of force development that are likely to be more akin to those observed during daily activities.

## CONCLUSIONS

When a change in rate of force development at low rates is required, only changing the initial discharge behavior of a few individual motor units, as opposed to all motor units such as that which occurs for ballistic contractions, would be sufficient to accommodate for the wide range of movement tasks that FHB is required to perform. We found such patterns when the rate of force development was altered for the required task. The fact that most studies have focused on comparing slow to ballistic movement, and the disagreement in findings between these studies and the current study, highlights the need for more research into the discharge behavior of motor units in conditions with slower, perhaps more ecologically relevant contraction speeds. It also needs to be noted that there is a large underrepresentation in the literature of studies on the intrinsic foot muscles which may adopt different discharge behavior than other skeletal muscles with different functional requirements. Studying a range of muscles in this way may therefore provide useful information on how the central nervous system finds different solutions to effect changes in contraction speed and the rate of force development. The observed substantial variability between motor unit behavior with increasing rate of force development should be further studied in other muscles and for different non-ballistic rates of force development, in order to fully probe the mechanisms behind this discharge behavior.

## ACKNOWLEDGEMENTS

The authors are grateful to all the participants who voluntarily underwent multiple fine-wire insertions in their foot. The authors also thank Dr Aurélie Sarcher for her assistance with the statistical analyses.

### Funding
J. Aeles is supported by a Marie Skłodowska-Curie Actions Individual Fellowship funded by the European Union (I-MUSCLE, 101063675). The funders had no role in study design, data collection and analysis, decision to publish, or preparation of the manuscript.

### Grant Disclosures
The following grant information was disclosed by the authors:
Marie Skłodowska-Curie Actions Individual Fellowship:  I-MUSCLE, 101063675.

### Competing Interests
The authors declare there are no competing interests.

### Author Contributions
- Jeroen Aeles conceived and designed the experiments, performed the experiments, analyzed the data, prepared figures and/or tables, authored or reviewed drafts of the article, and approved the final draft.
- Luke A. Kelly conceived and designed the experiments, performed the experiments, authored or reviewed drafts of the article, and approved the final draft.
- Andrew G Cresswell conceived and designed the experiments, performed the experiments, authored or reviewed drafts of the article, and approved the final draft.

### Human Ethics
The following information was supplied relating to ethical approvals (i.e., approving body and any reference numbers):

Human Research Ethics Committee of The University of Queensland approved the study (2018000460).

### Data Availability
The data are available at Figshare: Aeles, Jeroen; KELLY, LUKE; G. Cresswell, Andrew (2021): Motor unit data for rate of force development changes discharge behaviour. figshare. Dataset. https://doi.org/10.6084/m9.figshare.14745648.v3.

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
