# Peer review of "Flexor hallucis brevis motor unit behavior in response to moderate increases in rate of force development"

_PeerJ, doi:10.7717/peerj.14341_

## Round 0.1 · original submission · Major Revisions

We have obtained detailed comments from 3 reviewers, which are generally supportive and rather constructive, but some key criticisms are advanced. Most importantly perhaps, the validity of the statistical analyses has been challenged. Further analysis/data or other justification may be required. Please attend carefully to this and other issues noted by the reviewers in your revision. Thank you.

Reviewer 1 ·

Basic reporting

Given that the abstract length is half of the proposed limit by PeerJ (250/500 words), it would be beneficial if there was more information in the abstract, such as statistical outputs for each dependent variable. A clear separation between the results and the conclusion would also be beneficial, while adding a separate section for the conclusion itself.

In the introduction, it would be good if there was more background provided to the readers regarding the neural determinants to rapidly produce force. It is mentioned (Lines 55 and 56) that collectively, a decrease of recruitment threshold and an increase of discharge frequency contributes to an increase of rate of force development. This is an oversimplification of the phenomenon, without telling the readers if different neural factors are equally important or not. Previous research (Dideriksen, Del Vechio & Farina, 2019) used extensive simulations to suggest that the most important determinant across individuals is the recruitment speed of MNs. Recruitment speed could be estimated by computing the average time interval between the recruitment of identified MUs. Given the limitations of the method used in this study (intramuscular EMG with a very limited sample of MNs being identified), it would be challenging to estimate such measure. Nonetheless, it should be mentioned that the rate of recruitment of MNs highly drives changes in RFD.

According to PeerJ guidelines, the name of all authors should appear in the in-text references if they are 3 authors or less.

Experimental design

Participants’ characteristics and the number of participants are exactly the same than those reported in a previous study by the authors in the Journal of Neurophysiology (Aeles et al, 2020). It seems that these are not new experiments and therefore it is of paramount importance that readers are aware of this. This information should not be omitted. This should be made clear in the aims of the study in the introduction (especially because the aforementioned study is actually referenced by the authors at the end of the section), as well as in the methods section. Moreover, readers should know if participants performed additional procedures before, or in between the procedures that are currently described in the manuscript (e.g. procedures that were reported in the article published in the Journal of Neurophysiology).

Were participants asked to abstain from caffeine, alcohol, or physical exercise before the sessions?

Line 85: It is not clear why 5 participants collected data in a single session, while the other 5 were tested over multiple sessions. The underlying reason should be described.

Line 89: If the study was not registered in a database, this should be mentioned when the study is stated to be performed in accordance with the Declaration of Helsinki as pre-registration is a requirement of the current version.

Lines 97-99: Was there any digital filter applied?

Lines 103-105: The information provided here that participants were familiarized with the protocol over a 4-week period prior to the data collection sessions is surprising. How many sessions were conducted after all? This should be mentioned at the start of the methods section. Do all these sessions have to do with the fact that this is part of a larger study, with previously published data?

Lines 111-112: It would be helpful it there was information about what participants were instructed to do when they reached the peak value of the ramp contractions. For example, were participants asked to perform a trapezoidal contraction and only data from the ascending phase was analysed in this manuscript?

Lines 114-115: Was the order of the ramps quasi-randomised, to ensure that there was an equal number of participants starting with slow vs fast ramps?

Line 116: It is important to tell the reader how many ramps were used for data analysis. According to the abstract, it seems that it was 1 per condition? If so, which criteria were used to select the ramp for further analysis? If more than one ramp was used for further analysis, how was the data handled if the same MU could be observed across different trials of the same condition (i.e. slow or fast)? What is a set? Did participants performed first one type of contractions and then the other (i.e. one slow set and then a fast set, or vice-versa)? Or does a set include both a certain number of slow ramps and a certain number of fast ramps? This description should be more clear and a diagram with the procedures would be extremely useful.

Lines 122-124: It would be great if the ultrasound guidance could be better described. Was the 25-gauge needle used as a guide to see the correct angle and the depth for insertion of fine-wire electrodes?

Line 127: Information about the inter-electrode distance of the bipolar channels of intramuscular EMG data is missing.

Line 141-143: Was the decomposition of the data on two different occasions conducted by the same researcher or by different researchers?

Validity of the findings

To my understanding, the observations included in the paired t-test are the individual motor units that were identified. Unfortunately, I could not access the raw data used for statistical analysis, given that the file shared by the authors is a MATLAB file which is difficult to read (I highly suggest that raw data is presented in a more accessible format, such as Excel). In any case, if my interpretation is correct, it means that we will have participants with 1, 2, 3, 4, 5 or 6 observations (according to the individual MU data in Figure 3). This is a clear violation of an important assumption of a paired t-test: non-independence of observations. Thus, the statistical approach to answer the main research question of the present study is not suitable. Some observations will belong to the same participant and therefore they will not be independent of one another and cannot be merged in the same t-test. This is particularly concerning with this type of data, because it has been shown that motor unit firing data are correlated within a participant even across testing days (Tenan, Marti & Griffin, 2014). Thus, inappropriate conclusions can be inferred if all identified motor units are “put in the same bag” with a ordinary least squares-based test (ANOVA, t-test, OLS regression, etc). Failing to account for correlated data in a paired t-test can substantially inflate type 1 error due to the artificially low standard errors. Ideally, the whole sample of motor units could have been used in the statistical analysis if a repeated measures nested linear mixed-effects model was used (e.g.: Boccia et al., 2019). Another alternative would be to get an average score per participant and conduct a t-test. Although this latter approach would not violate any statistical assumptions, it would have important limitations: less statistical power, data would be averaged out and certain participants would have a raw value, while others would have an averaged value, when conducting the t-test.

Moreover, a “linear correlation coefficient” (I assume Pearson’s correlation?) was used with the whole sample of test units (Figure 4). This is statistically inappropriate for the same reason. There is non-independence among certain data-points which is an important violation of this correlation analysis. In this case, those participants who have more observations (i.e. more MUs identified) will contribute more heavily to the final R value. Another method has to be used which accounts for the non-independence of observations, such as the repeated measures correlation (Bakdash & Marusich, 2017).

Then, it can be concluded that the interpretation of the data in this study is not statistically sound. Therefore, I was not able to comment on the results or discussion.

Additional comments

Not applicable.

Reviewer 2 ·

Basic reporting

This appears to be an extension of a recent paper by this group in the same experimental setup.
The challenge as presented at this stage is appreciating the rationale and question as applied to this intrinsic foot muscle. Several statements and perhaps inconsistencies lead me to this lack of appreciation.
1) Lines 43/44 indicate we know a great deal about foot muscle functional adaptability, but lines 63/64 indicate we know very little about MU behaviour leading to a ‘need’ (line 58) to study MU behaviour in relation to RTD. This is confusing to follow. Why is this muscle special or suitable for this ‘need’? Authors go on to state (lines 59-61) that foot muscles are ideal because it is implied for some unique reason foot muscles produce forces across a wide range (?). Indeed most all limb muscles do this. In lines 72-74 it is stated that this intrinsic foot muscle behaves remarkably differently than other similarly-sized muscles. Intrinsic muscles of the hand are different and are a similar size, but this is not apparently considered. What is this ‘distinctive’ force-control mechanism for FHB? Those features should be described here to help perhaps convince the readership of the value or utility of this muscle. It seems FHB follows recruitment and rate coding features similarly to other spinally innervated muscles, but perhaps not, or I am missing something in all of this.

2) To explore the RTD idea of not fast but not too slow, why this muscle? The challenge for many people with this muscle is that purposefully, without training it seems, it is not easy to regulate RTD that is moderate. Some other muscles might seem much more appropriate (?). The value of exploring RTD with MU behaviours in general is not well articulated or why with this specific ‘distinctive’ muscle. Line 75 refers to the FHB as a ‘significant’ foot muscle. Please identify which foot muscles are insignificant. All this creates a lack of rationale and purpose for the paper and requires much improvement if possible.

3) That it apparently requires weeks of training, is the test group then representative of a trained neuromuscular system? - which from other studies has been shown to induce MU behaviour changes. This concept should be incorporated into any interpretation and make limit the value of this muscle being ‘distinctive’. Likely more details are required to describe the training regimen.

4) With the description of this foot force measuring device it is not clear how activation of the plantar flexors in addition to FHL was controlled or monitored. It is unclear how useful surface would be over FHL and FHB - please see: https://www.frontiersin.org/articles/10.3389/fphys.2019.01283/full
How did you establish peak or MVC values for each of these muscles to normalize the surface EMG for the assessment of co-activation? It might be helpful to show representative tracings of these individual MVCs with surface EMG. Why not also monitor PFs as potential contributors to ankle plantar flexion which seems would impact the net force in this device?
5) Line 207 – please explain ‘highly distinctive’ in this analysis.

6) With a variety of spikes per participant and per condition the MUs should properly be analyzed using linear regression (Tenan M.S. et al 2014 JEMGK). Power seems low as well Please update the analyses.

7) Lines 224-226 I do not think a weak correlation can allow this conclusion. Please be careful not to assign cause and effect.

8) I do not follow how these conclusions on lines 244-246 have been derived from these data. I expect all motor unit systems have a high degree of flexibility in force production and are task-dependent. What is new or special about these results in this muscle? One could as well argue that FHB is not nearly so important for forward propulsion as FHL and may simply be more of a foot stabilizer rather than a main effector for walking. Perhaps because it is indeed as you state relatively small that MU behaviour as a foot stabilizer over some moderate changes in RTD in relation to the whole range available is not at all surprising and has little to do with anything particularly special or unique about FHB. Do the same experiment in TA or a hand muscle before assigning a particular outcome as unique.

9) Indeed the it seems these results are not that unique (lines 250-258) and that perhaps the smaller magnitudes of changes in this study are due simply to the smaller ranges in RTD employed (?) This all needs a clearer re-evaluation.

10) Lines 289-290 One could also argue that the tasks are not similar – 2 fold difference in RTD

11) Much of the discussion is speculation or summary of concepts well-appreciated from other studies, leaving this reviewer uncertain of the value and utility of these results keeping in mind the admitted limitations and others noted in this review.

12) Lines 313-315 It is not clear that this muscle has really displayed unique or remarkably different strategies as compared with any other spinally innervated limb muscle.

13) Fig 2 could benefit from some overlays or the repetitive firings of each identified MU train.

14) Fig 3 Seems to show that RTs and changes in DR are variable and not entirely convincing with this sample size and perhaps not analysed properly to be very convinced that there is any systematic change or differences.

Experimental design

I have captured these 3 sections/questions all within the first heading. It should be obvious where concerns are raised in background, design and validity and many are intermingled.

Validity of the findings

Please see above

Additional comments

For section 1 my main concern or what I would consider a failure as presented is unclear rationale leading to the hypothesis and the lack of improved understanding about MU behaviour and this muscle model

For Section 2 I question the research question as presented again using this model with limitations in data analyses

For section 3 I remained unclear of the value and utility of the results with improving understanding of MU behaviour and why this muscle

Thus, according to the directions for review, there are 'fails' in each of these 3 sections. Whether with major revisions it could be all improved I do not know but I seem constrained by the guidelines in my choice.

·

Basic reporting

This manuscript is well-written, clear and concise.

Experimental design

The experimental design rigorous and is relevant for the proposed research questions. The research question is well defined and most of the methods are well described. I do have comments for clarification that may impact the interpretation of the results. Please see the additional comments for details.

Validity of the findings

No Comment.

Additional comments

The submitted study investigated whether MU recruitment and rate coding was altered for the FHB during two different rates of force production of an intrinsic foot toe flexor. There was no difference in motor unit recruitment threshold, but a small difference in initial discharge rates for the different rates of force production. RMS amplitude was also not different between contraction rates. The intrinsic foot muscles are understudied, but an important avenue for research regarding the neuromechanical control of movement. The authors have submitted a well-written, concise and clear document that provides novel insights on the control of human movement. The methodologies seem rigorous and the experimental design is suitable for the research questions proposed. This is a well-thought out study and I only have minor questions and comments to help improve clarity of the manuscript and potentially help strengthen the presentation and interpretation of the results. I have included my general and specific comments below.

General Comments
The authors report EMG amplitudes representing the FHB and FHL to account for activity from other potential muscle activity contributing to the toe flexion task. Because the Abductor Hallucis can also contribute to toe flexion, was EMG monitored from that intrinsic foot muscle? Is it possible that variability within the activity of the AH could contribute to the variability in MU behavior reported here? Please include if the data is available. If not, please comment on AH activity and its applicability to the experimental design and interpretation of results in this study.

Because the ramp contractions performed in this study are based on an MVC, it would likely benefit the manuscript if the authors could provide greater details on MVC procedures (e.g., number of attempts, duration of attempt, etc.). Clarification on the number and duration of MVCs performed would help for replication of the study design.

All participants in this study essentially performed an extensive training regime of the toe flexors. Would this training have any implications on the MU behavior outcomes reported here? I understand that the muscle group may be difficult to isolate and training may be necessary, but would the training have altered the MU behavior of a typical ramp contraction? Could training explain some of the findings between the rates of contractions? Would naïve participants use a different MU strategy? This may be an important note to include.

Specific Comments
Introduction Lines 63-64. “Our knowledge on the motor unit behaviour of the intrinsic foot muscles during dynamic tasks is somewhat limited.” It is not clear how work or, in this case, limited investigations on dynamic tasks is important for the rationale of this paragraph and the introduction. The authors are focusing on isometric contractions. Please clarify.

Introduction Lines 71-74. “Our recent work on the flexor hallucis brevis (FHB) muscle revealed unique motor unit discharge behaviour compared to similarly-sized muscles, suggesting that motor units in the intrinsic foot muscles may have a distinctive force-control mechanism (Aeles et al. 2020)." It may be relevant to clarify what the authors mean by unique behavior to provide the reader a better understanding of the interesting neural control strategies of the intrinsic foot muscles. In a similar regard, why did the authors choose the FHB over other intrinsic foot muscles? It may be beneficial to develop a brief rationale as to why the FHB is the choice model over other intrinsic foot muscles.

Methods Line 102. “The knee and hip joint were positioned at approximately 90o of flexion…” The degrees symbol is not positioned accurately.

Methods Lines 103-105. “All participants were extensively trained to perform the submaximal MTP flexion task at two different ramp rates over a four-week period prior to the data collection.” I understand that familiarization is necessary for the isometric task performed with the intrinsic foot muscle used here. Essentially the experiment is a training study without any baseline data. I’m wondering if the motor unit behavior characteristics observed here are owing to a training effect over the four-week period? The authors may want to acknowledge this contribution within the manuscript for clarification.

Methods Line 107. “All participants performed a standardised warm-up before all sessions.” What is a standardized warm-up? What activities were included in the statndardized warm-up? Please include.

Results Lines 214-222. It may be beneficial to visualize some of the MU characteristic data as means and variability as well as including individual values in a figure to have a better understanding of the spread in the MU data (e.g., MU recruitment threshold, DR characteristics).

Figure 1B. It may be of interest to also include the absolute force values for the ramp contraction to provide a better comparison to other muscle groups and studies.

Figure 2. It may be helpful for the reader if the authors provide greater details regarding the MU characteristics. For example, reporting the MU RTs on the figure, as well as potentially firing rate variability and initial and mean firing rates of the units depicted in the figure.

---

## Round 0.2 · Major Revisions

We have two reviews for the paper; sorry for some delays. One prior reviewer was unavailable and we needed a second review. Unfortunately that review (Reviewer 4) is quite critical. Together with the re-review by Reviewer 1, this study needs a re-think and at least moderate revisions if not very major ones, and if resubmitted it must win the reviewers over more.

Reviewer 1 ·

Basic reporting

Abstract:
It would be beneficial if the aim would be more explicit in the abstract. The last two sentences could be rephrased. For example, it would be more important to note that the rate of contractions assessed in this study are more relevant for activities of daily living. The fact that MUs are recruited across a wide range of recruitment thresholds might not be relevant for the abstract and some readers might not understand what “does not span the longitudinal arch” mean and its implications.

The number and sex of participants should be included.

Line 27: Electromyograms (global EMG) were decomposed, which allowed the identification of MU action potentials from individual units. So, saying “we decomposed motor unit action potentials” sounds odd.

Line 36: For clarity, the authors could add the word “initial” before “discharge rate”.

Line 43: Stating that increases were observed during a modest range is not exactly specific to these findings. For example, increases in RFD from very fast to ballistic could still be seen as a “modest range”. For clarity, this should be rephrased.

Line 46-47: It is not clear that the authors are talking about data from previous studies, as some readers who read the abstract might initially think that the authors also looked at changes during ballistic contractions in other muscle groups.

Lines 48-49: As I express in another comment related to the discussion of the findings, I do not think that the fact that MUs responded differently is evidence of a flexible system.


Introduction:

Lines 63-66: This sounds odd and I don’t understand what the authors mean here. Moreover, in this reference it was shown that EMG amplitude during a fatiguing contraction cannot be used as a surrogate of neural drive, which is not directly relevant to the present study.

The flow of the 2nd paragraph of the introduction could be improved, and some important details are missing. The first sequence of studies is related with comparisons between slow and ballistic contractions. Whereas the new information regarding the findings by Dideriksen et al. (2020) are in respect to maximal rate of force development during explosive contractions. Thus, it would be useful if these details would be added for clarity and the sentence in lines also 80-84 rephrased.

Line 98: It would be beneficial if the authors would describe the meaning of “does not span the longitudinal arc”. Does this mean that the influence of other synergists is less likely and the interpretation of MU parameters during a ramp contraction less complex?

Line 109-111: I assume that the authors mean that the modulation of MU behaviour between different, but moderate, rates of contraction is unknown. However, this sentence sounds odd as the motoneurons drive muscle contractions. So the cause-effect relationship sounds reversed.


Methods:

I feel that the flow of the methods could be improved. It could flow better if the “Participants” paragraph was solely about participants’ characteristics and ethics approval. Then at the beginning of the next sub-section (or while creating a new one with an “experimental overview”), it could be fully described how many sessions participants attended as well as the full description of the training period.

Lines 183-184: It seems that the new sentence should come after the following sentence to flow better.

Statistics section: Which software was used for each analysis? If analyses were conducted in R, using RStudio environment, packages should be mentioned too.


Results:

After all, how many MUs included in the analysis? 19, 21 or 33? It is not clear by reading the first paragraph in the results section and counting the number of observations in Figure 3.

Lines 328-329: These ranges are about all participants. Readers might wonder low-threshold units were identified in some participants and high-threshold units in other participants, for example. At the moment, readers can only see the example in Figure 2. It would be useful if in Figure 3 or 4, another panel showing the absolute (not change) of recruitment thresholds for the individual units. I was wondering if this information was at least in the supplementary files, but the Excel file solely presents the raw data of timings of firing events, to my understanding.

Related to the previous comment, why very low-threshold MUs were not identified (i.e., the ones recruited at onset of contraction)? Does it have to do with the fact that very low-threshold MUs have a smaller action potential and the decomposition is biased towards bigger units? It would be beneficial if this was contemplated in the discussion.

Lines 331-332: If the authors would like to provide information to the readers about the magnitude of relative increase, they could also consider mentioning that the initial discharge rate was 10.2% higher in the fast ramps.

Caption of Figure 2: The yellow rectangles are not really “transparent”, so this word could be removed.


Discussion:
Lines 354-355: Here and elsewhere, it would be useful for the reader if it was explicit what was the rate of contraction in each task in % MVC per second.


Conclusion
Lines 443-447: It would be useful for the readers if these highly speculative sentences at the start of the conclusion were replaced by the main findings of the study.

Experimental design

Were participants always tested at the same time of the day? Whether or not this happened should be explicit.

I assume the different testing days have to do with the inability in some participants to get enough MUs in one single session? This could be briefly mentioned. What was the criterion to stop trying to find more MUs?

Line 167-169: How much of an increase exactly? Does this refer to an increase from the MVC to 125% MVC?

Lines 169-171: It would be beneficial to provide an average of sessions per participant. Moreover, it could be described how many required 1 session, 2 sessions, and more than X number of sessions over a period of 4 weeks. Although it is mentioned in the caption of one figure, it could also be mentioned here that data from the training is not presented in the manuscript.

Line 170-171: What were exactly the criteria to consider that one participant was well-trained and could start testing? Was the criterion absence of EMG of FHL? Was this accomplished by visual inspection and subjective assessment or by objective criteria?

Lines 175-176: How many ramps and what was the duration of each ramp?

Line 189: Do authors mean quasi-randomised “between sets”. Does this mean that some participants started with a slow and others with a fast contraction and then an alternated fashion was followed until a sufficient number of sets were performed? This could be more explicit, for clarity.

Line 192: How many sets? Average and range of number of sets could be given.

Line 200: “different motor unit” or “different set of motor units”?

Lines: 225-227: I assume a ground/reference electrode was used and this should be described here in the methods. I can see something in the malleoli in Figure 1A – was this the ground/reference electrodes?

297-299: It is not clear why the authors looked at repeated measures correlations between changes in mean DR and changes in RT and not between changes in other dependent variables. If the authors examined other correlations which proved not to be significant, they should also report this.

It is not explicit how the initial and final DR were calculated, as the first mention of these variables occurs in the results section. I am assuming the instantaneous firing frequency of the first two spikes of the MU for initial DR and instantaneous firing frequency of the last two spikes of the MU for the final DR. This information should be incorporated in the methods.

Moreover, it would be useful if it was described how recruitment thresholds were calculated (estimated?) since, in a response to reviewer 3, the authors mentioned that force output was measured in voltage.

Validity of the findings

I was happy to see that the authors reanalysed the data with the tests that I proposed. I believed that this approach is much more suitable for this design, and improved the quality of the paper. As a follow-up, it is not specified whether estimated marginal means were used in post-hoc tests? In results, I can see means and wonder if these are arithmetic (or observed) means or estimated marginal means. As a nested linear mixed model was used, estimated marginal means should be reported – they are estimated from the model, rather than an average of the data values. An estimated marginal mean would be the same than an arithmetic mean if the design is balanced, no missing values and if there was no adjustment for covariates or random effects. I’d suggest the package emmeans (Length & Length, 2018).

In this study, the repeated measures correlation was conducted to examine if there was an association between changes in two different dependent variables. I believe the authors mean this, but in the current version (in the abstract and in the methods) it sounds like this statistical approach was used to investigate differences (i.e., cause-effect, rather than correlation).

The fact that caffeine ingestion was not controlled is a limitation of the current project and it should be acknowledged in the discussion. It could contribute to variability within day (i.e., given the pharmacokinetics of caffeine, the magnitude of caffeine effects in the neuromuscular system could vary during the session) and between days (if timing and quantity of ingestion was different).

Lines 352-354: “We also found no increase in surface EMG amplitude with faster contractions, which suggests that there was no significant increase in the number of motor units recruited.”. This information described here and in lines 403-405 should be revised. Changes (or absence of changes) in sEMG amplitude between two tasks with different RFDs cannot be used to examine whether a different number of motor units were actively contributing to the contraction in this muscle. See the section “Motor Unit Recruitment and Rate Coding” in Vigotsky et al. (2018) “Interpreting Signal Amplitudes in Surface Electromyography Studies in Sport and Rehabilitation Sciences”.

Lines 356-361: I agree with comment 8 of Reviewer 2, from the 1st round of reviews. I also do not think a considerably high variability in the outcome variables, when considering changes from slow to fast ramps, necessarily means a “greater degree of flexibility in force production”, and I feel that it should not be a conclusion of these findings. Changes in these variables are necessarily related to the changes in the balance of excitatory and inhibitory ionotropic synaptic inputs, neuromodulation, intrinsic excitability of motoneurons and spike thresholds. Biophysically speaking, I cannot understand the evidence (and the advantageous nature) of a “greater degree of flexibility”. It was great that the authors presented individual data from each MU in Figure 3, in a creative manner. Nonetheless, I am afraid this variability could be due to other factors:
- It could be related with the fact that only a limited number of MUs were identified (in some participants only one MU; there was a small portion identified of the total number of MUs actively contributing to the contraction, as mentioned by the authors) and who knows if there was a differential effect across the motoneuron pool. This could mean that in some participants it could have been identified MUs which observed an evident change in one direction, but MUs which exhibited changes in a different direction or an absence of change in MUs from other participants. Differences across the motoneuron pool could be due to differences across motoneuron size or due to the inter-dependence of MU parameters in this task: since participants were asked to follow a force trace, what happened to the first recruited MUs [and note that very low-threshold MUs were not identified] will influence the behavior of the subsequently recruited MUs.
- Even though units within the same set were used (as explicit in lines 243-246), it is important to note that different sets within days and between days were used to answer the research question. Confounding factors such as fatigue, warm-up of contractile properties, warm-up of persistent inward currents in motoneurons, changes in neuromodulatory inputs due to different levels of arousal in different time-points during a session could contribute to the variability of the measures. In respect to variability within-day, even though rest was provided between sets, variability could still occur. Moreover, as there is data on MVCs, it could be beneficial to indicate whether there was an effect of time in MVCs, denoting absence (or presence?) of potentiation or fatigue (lines 197-198). Nonetheless, even if there’s no effect of time, the contamination of the aforementioned confounding factors could still occur. I assume (not explicit in the manuscript; it would be beneficial if it was more clear) that the performance of multiple sets and in different days (for most participants) was necessary to get a sufficient number of MUs, but the related limitations should be acknowledged.
- Possible variability between sessions because different days were used in most participants (e.g., different caffeine ingestions, state of arousals, sleep or different time of days)
- Possible differential contribution of synergist muscles between slow and faster contractions (authors mention abductor hallucis in the discussion and mention that pilot data suggest that contribution of plantar flexors was negligible when responding to reviewer 2; the latter could possibly be added to the discussion), and these modulations could vary between participants.
It would be very useful for the readers, if the authors would reflect about these factors and mention them together in the discussion as possible confounding factors for the evident “variability”. The variability could be a natural consequence of any of these and not necessarily evidence of an advantageous “flexibility”, which is very speculative and not supported by the data.

Lines 362-375: Throughout the discussion of the findings related to initial DR in this paragraph or in the following paragraph, it is not mentioned the underlying mechanism(s) influencing initial MN DR. This measure should be associated with the strength of the injected current. It makes sense that the net excitatory synaptic input is higher in the ramps that were relatively faster in the current study, but the magnitude of difference should not be as high as it is when we compare slow vs ballistic. If the authors agree with my interpretation, they could then further speculate the “relative importance” of initial discharge rate in the type of tasks performed in this study (higher?) vs. in the maximal rate of force production (lower) (Dideriksen et al., 2020), at the end of the following paragraph.

Lines 369-371: The “secondary drop in DR rate” the authors mentioned is a non-linear decrease in firing rate which is partly influenced by spike frequency adaptation, associated with the inactivation of sodium conductance. This happens in explosive contractions and the drop is quantified as the decrease of firing rate after the very initial firings at force onset. Given that the authors did not identify any decrease in firing rate (which probably didn’t occur in the ascending phase as these were relatively slow ramp contractions) and as very low-threshold MUs were not identified in this study, this interpretation cannot be made.

As discharge frequencies higher than 50 Hz were discarded, authors could not examine the occurrence of doublets. Do authors think that the occurrence of doublets between slow and not so slow ramps would be different? This could also be mentioned.

·

Basic reporting

No comment

Experimental design

No comment

Validity of the findings

The study compared the recruitment thresholds and discharge frequencies of single motor units in an intrinsic foot muscle (FHB) during a toe flexion task that involved isometric contractions at two rates of force development. Based on the findings reported by other groups, the authors hypothesized that recruitment threshold would decrease and discharge rate would increase with an increase in the rate of force development. However, the authors expected the discharge behavior of motor units in FHB to differ from previous results due to the unique functional demands on this muscle.

The main findings of the study that discharge rate at recruitment did indeed increase as the rate of force development increased from a slow to a faster speed. In contrast, there was no change in the recruitment threshold force across contraction speeds. The authors conclude that the discharge rate of recruited motor units is prioritized over lowering recruitment threshold to achieve a faster rate of force development. The authors do not appear to appreciate the difference in the underlying mechanisms for these two features of motor unit activity. Although, the modulation of discharge rate reflects the integration of synaptic input with the intrinsic properties of the motor neurons, recruitment threshold is determined by the electromechanical delay between the discharge of action potentials and when the force contributed by the motor unit is needed to achieve the prescribed force trajectory. The finding of no difference in recruitment threshold for the two rates of force development suggests simply that the electromechanical delay did not differ between these two conditions. It does not suggest a difference in discharge behavior between FHB and other muscles.

Additional comments

No comment

---

## Round 0.3 · Minor Revisions

We obtained one review that appraises the revised manuscript as a substantial improvement, but there is still some over-reaching in parts of the manuscript and it must be made clearer, especially in the Conclusion, what is truly supported by the data vs. a speculation from the data.

Reviewer 1 ·

Basic reporting

Some typos/grammar mistakes should be corrected before final publication.

Experimental design

No comment.

Validity of the findings

The authors have provided a thoughtful revision. Nonetheless, I would like to still point out that the authors continue to not offer convincing evidence that the considerably high variability in the outcome variables is an advantageous characteristic of a flexible motor pool. This variability can be easily explained by several methodological and physiological factors (as now more evidently pointed out in the discussion), or by the possibility that an increase in the initial discharge rate in most MUs was sufficient to accommodate the increase in rate of force development. It would be more beneficial for the readers if this would be more evident in the conclusion of the abstract and in the Conclusion section of the manuscript.

Suggestion for improvements: The increase in "initial" discharge rate should be more evident in the conclusion of the abstract and mentioned in the conclusion of the manuscript. The link between the variability of the outcomes and the "flexibility" of the system could be more speculative in the final Conclusion section of the manuscript.

Additional comments

No comment

---

## Round 0.4 · accepted · Accept

I have checked the revised manuscript and the changes are satisfactory. Thank you for your efforts, and congratulations on having the paper accepted for publication!